# Global Methylome Scores Correlate with Histological Subtypes of Colorectal Carcinoma and Show Different Associations with Common Clinical and Molecular Features

**DOI:** 10.3390/cancers13205165

**Published:** 2021-10-14

**Authors:** María del Carmen Turpín-Sevilla, Fernando Pérez-Sanz, José García-Solano, Patricia Sebastián-León, Javier Trujillo-Santos, Pablo Carbonell, Eduardo Estrada, Anne Tuomisto, Irene Herruzo, Lochlan J. Fennell, Markus J. Mäkinen, Edith Rodríguez-Braun, Vicki L. J. Whitehall, Ana Conesa, Pablo Conesa-Zamora

**Affiliations:** 1Facultad de Medicina, Universidad Francisco de Vitoria, Ctra. Pozuelo-Majadahonda, Km 1800, Pozuelo de Alarcón, 28223 Madrid, Spain; mc.turpin.prof@ufv.es (M.d.C.T.-S.); i.herruzo.prof@ufv.es (I.H.); 2Biomedical Informatics & Bioinformatics Platform, Institute for Biomedical Research of Murcia (IMIB)/Foundation for Healthcare Training & Research of the Region of Murcia (FFIS), Calle Luis Fontes Pagán 9, 30003 Murcia, Spain; fernando.perez8@um.es; 3Department of Pathology, Santa Lucía General University Hospital (HGUSL), C/Mezquita s/n, 30202 Cartagena, Spain; jgarcia652@ucam.edu; 4Facultad de Ciencias de la Salud, Universidad Católica de Murcia (UCAM), Campus Los Jerónimos, 30107 Guadalupe, Spain; 5Group of Molecular Pathology and Pharmacogenetics, Institute for Biomedical Research from Murcia (IMIB), HGUSL, 30202 Cartagena, Spain; 6IVI Foundation, Instituto de Investigación Sanitaria La Fe (IIS La Fe), 46026 Valencia, Spain; patricia.sebastian@ivirma.com; 7Department of Internal Medicine, Santa Lucía General University Hospital (HGUSL), C/Mezquita s/n, 30202 Cartagena, Spain; javier.trujillosantos@gmail.com; 8Biochemistry and Clinical Genetic Center, Virgen de la Arrixaca University Hospital, 30100 Murcia, Spain; pablo.carbonell@carm.es; 9Department of Social Psychology and Methodology, Universidad Autónoma de Madrid, 28049 Madrid, Spain; eduardo.estrada@uam.es; 10Cancer and Translational Medicine Research Unit, Department of Pathology, University of Oulu, P.O. Box 5000, 90014 Oulu, Finland; anne.tuomisto@oulu.fi (A.T.); markus.makinen@oulu.fi (M.J.M.); 11QIMR Berghofer Medical Research Institute, Herston, QLD 4006, Australia; lochlan.fennell@qimrberghofer.edu.au (L.J.F.); Vicki.Whitehall@qimrberghofer.edu.au (V.L.J.W.); 12Conjoint Internal Medicine Laboratory, Pathology Queensland, Herston, QLD 4006, Australia; 13Faculty of Medicine, The University of Queensland, Herston, QLD 4072, Australia; 14Clinical Oncology Department, Santa Lucía General University Hospital (HGUSL). C/Mezquita s/n, 30202 Cartagena, Spain; edith.rodriguez.braun@gmail.com; 15Microbiology and Cell Sciences Department, Institute for Food and Agricultural Sciences, Genetics Institute, University of Florida, Gainesville, FL 32611, USA; vickycoce@gmail.com; 16Department of Clinical Analysis, Santa Lucía General University Hospital (HGUSL), C/Mezquita s/n, 30202 Cartagena, Spain

**Keywords:** colorectal carcinoma, serrated pathway, microsatellite instability, serrated adenocarcinoma, CIMP, DNA methylation, epigenetics of carcinogenesis, methylome, conventional carcinoma, methylation score

## Abstract

**Simple Summary:**

Aberrant patterns of methylation at specific genome sequences (CpG) are driving forces for the carcinogenesis process of different cancers, including colorectal cancer (CRC) and, typically, include gene promoter hypermethylation and global genome hypomethylation. Despite that CRC is diagnosed based on histological evaluation, its association with global methylation patterns has not been established so far. By studying the methylation status of 450,000 CpG sites in 117 colorectal specimens, the authors obtained global scores reflecting the methylation level at different sequence distances from the genes. The authors identified that histological CRC subtypes show different global methylation scores and that the shift from promoter hypermethylation to genomic hypomethylation occurs at a small sequence between 250 bp and 1 Kb from the gene TSS.

**Abstract:**

Background. The typical methylation patterns associated with cancer are hypermethylation at gene promoters and global genome hypomethylation. Aberrant CpG island hypermethylation at promoter regions and global genome hypomethylation have not been associated with histological colorectal carcinomas (CRC) subsets. Using Illumina’s 450 k Infinium Human Methylation beadchip, the methylome of 82 CRCs were analyzed, comprising different histological subtypes: 40 serrated adenocarcinomas (SAC), 32 conventional carcinomas (CC) and 10 CRCs showing histological and molecular features of microsatellite instability (hmMSI-H), and, additionally, 35 normal adjacent mucosae. Scores reflecting the overall methylation at 250 bp, 1 kb and 2 kb from the transcription starting site (TSS) were studied. Results. SAC has an intermediate methylation pattern between CC and hmMSI-H for the three genome locations. In addition, the shift from promoter hypermethylation to genomic hypomethylation occurs at a small sequence between 250 bp and 1 Kb from the gene TSS, and an asymmetric distribution of methylation was observed between both sides of the CpG islands (N vs. S shores). Conclusion. These findings show that different histological subtypes of CRC have a particular global methylation pattern depending on sequence distance to TSS and highlight the so far underestimated importance of CpGs aberrantly hypomethylated in the clinical phenotype of CRCs.

## 1. Introduction

Epigenetic changes modify gene expression without altering DNA sequence, this regulation being heritable and reversible [1]. DNA methylation is an important epigenetic modification and mainly affects CpGs islands, which are regions of DNA spanning at least 200 bp and have a GC content of >50% [2] CpG islands are often located at gene promoters, and methylation of these islands can alter gene expression.

One of the earliest events in colorectal carcinogenesis, present even at the stage of aberrant crypt foci [3], is focal hypermethylation of gene promoters and, global hypomethylation of DNA [4]. Promoter hypermethylation suppresses gene expression, and mainly affects tumor suppressor genes. Global hypomethylation may contribute to cancer development by inducing DNA recombination, hijacking transcriptional factors or activating the expression of oncogenes, lncRNAs or microRNAs, which are normally not expressed in healthy colon tissue [4]. Although the ENCODE project has determined that most of the non-coding DNA in the genome is likely playing a functional role [5], the mechanisms of local regulation by non-coding elements are largely unknown, as are the proximity criteria required to engage in regulatory relationships with target genes.

Aberrant DNA methylation in colorectal carcinomas (CRCs) is a global phenomenon that is currently assessed by measuring the methylation status of only a few genes by the so-called CpG island methylation phenotype (CIMP) test. CIMP is associated with proximally located sporadic CRC. These cancers typically display *MLH1* promoter methylation, *BRAF* mutation and high levels of microsatellite instability (MSI-H) [6,7]. Sporadic CRCs that methylate the *MLH1* promoter methylation and develop microsatellite instability (MSI) are considered as one end-point of the serrated neoplasia pathway. These cancers display distinct histological features [8]. Serrated neoplasia can also lead to the development of another subtype of CRC called the serrated adenocarcinoma (SAC). SAC is more frequently KRAS mutated and usually microsatellite stable [9,10]. Although some studies have analyzed differentially methylated genes in SAC, conventional carcinoma (CC) and hmMSI-H [11,12], the use of scores representing the genome methylation in different histological subtypes of CRC has not yet been explored. In this study, we analyze whether global CpG methylation scores at three different regions defined by their distances to the gene transcription starting site (TSS) are associated with particular clinical, histological and molecular features of CRC.

## 2. Materials and Methods

### 2.1. Patients and Tumor Samples

The clinical and pathological characteristics of the study patients were reported previously [13,14,15,16,17]. This study was approved by the Local Ethical Boards from participating hospitals (Ref: PI12-01232), in agreement with the ethical principles laid down in the 1964 Declaration of Helsinki. Informed consent was acquired from all participants. SACs and conventional carcinomas (CCs) were diagnosed based on recognized criteria [18,19]. The CC group comprises the WHO entities of adenocarcinoma NOS, mucinous and non-mucinous. No adenoma-like, micropapillary, adenosquamous adenocarcinomas, or signet ring cell, poorly cohesive, undifferentiated sarcomatoid carcinomas were included in our CC series [20]. Fresh frozen tissue from 40 SACs was collected from the Santa Lucia University Hospital, Cartagena Spain (*n* = 22) and the Oulu Hospital, Finland (*n* = 18) for DNA methylation analysis. We also assessed DNA methylation in randomly selected healthy mucosa adjacent to 15 SACs (11 Spanish and 4 Finnish cases). Frozen samples from 32 CCs matched with SAC for gender, age, TNM stage and WHO score, were selected as a control group from the same tumor banks (24 Spanish and 8 Finnish cases), these were accompanied by adjacent mucosal tissue from 14 participants (10 Spanish and 4 Finnish). In addition, a previous series of 10 CRCs [10,21] showing MSI-H molecular and histological features (mucinous, and medullary carcinoma, intra- and peritumoral infiltrating lymphocytes, “Crohn-like” inflammatory reaction, poor differentiation, tumor heterogeneity and “pushing” tumor border [8]) termed hmMSI-H and corresponding to the WHO medullary adenocarcinoma subtype [20], were also studied in conjunction with 6 normal adjacent mucosal samples. Clinicopathological features of the cases are shown in Table 1. Tumor budding was assessed as low when less than 10 tumor buds were identified and high-grade tumor budding when showing 10 or more tumor buds [14].

### 2.2. DNA Extraction

An average 10 mm^3^ tissue volume was excised from frozen specimens using a disposable sterile biopsy punch. DNA was extracted using the QiaAmp DNA Mini Kit (Qiagen, Hilden, Germany). In brief, a Tissueruptor (Qiagen, Hilden, Germany) was used to disrupt and homogenize the tissue in 100 µL ATL buffer, which was then incubated with 10 µL proteinase K (Ref: 19133) at 56 °C to ensure complete tissue lysis. DNA was extracted from lysates using the Qiacube automated extraction platform and the QiaAmp DNA Mini Kit (Ref: 51306).

### 2.3. Treatment with Bisulfite and DNA Methylation Assay

Genome-wide DNA methylation screening was assessed using HumanMethylation450K BeadChip assay (Ref: WG-314-1002) (Illumina, Inc, San Diego, CA, USA). This array investigates the cytosine methylation status of over 480,000 CpG sites [22]. Briefly, genomic DNA from each sample (~1000 ng) was bisulfite treated with EZ DNA Methylation Kit (Ref: D5002) (Zymo Research, Orange, CA, USA) as per the manufacturer’s instructions. Bisulfite-treated DNA was amplified at 37 °C for 20–24 h and the resulting DNA was fragmented by an enzymatic process, precipitated, resuspended, placed onto an Infinium Human Methylation 450 K BeadChip (Illumina, San Diego, CA, USA) and hybridized at 48 °C (16–24 h). The chip was fluorescently stained and imaged using an Illumina i-SCAN instrument. BeadArray data were analyzed by Illumina’s GenomeStudio program (Methylation Module) to assign site-specific DNA methylation β-values to each CpG site. High-throughput data from our series are fully accessible through Colportal (www.colportal.imib.es, accessed on 7 June 2021), an online integrated platform [16].

### 2.4. Analysis of Methylation Data

Data were analyzed in R (version 3.2.1, June, 2015) [23]. The methylumi [24] R package was used to read methylation data and remove probes with a low detection value (*p* < 0.01) in more than 95% of the samples, probes measuring SNPs and probes mapping to the sex chromosomes. The remaining probes underwent a three-step normalization procedure. First, the color bias adjustment included in the methylumi R-package was applied. Then, wateRmelon [25] R-package was used to perform quantile normalization between samples, where type I and type II backgrounds were equalized and then methylated and unmethylated intensities were quantile normalized separately. Finally, the BMIQ [26] intra-sample normalization procedure included in wateRmelon R-package was applied to correct the bias of type II probe values.

Three different regions of interest were defined as a function of the distance to the TSS of the associated gene: I (1 bp–250 bp), II (250 bp–1 Kb), III (1 kb–2 Kb). Probes were mapped to these regions, and the average methylation level of probes mapping to each region was used as a score of methylation in the region. Scores distributions were analyzed per sample disease status (tumor vs. normal), tumor type, demographics and clinical outcome. Average global methylation values were also obtained and analyzed according to their location with respect to CpG Islands (CpG Island, N_shore, S_shore, N_shelf, S_shelf, Open Sea). In addition, average methylation values were obtained for the N(N_shore + N_shelf) and S_(S_shore + S_shelf) regions.

These regions were defined based on the manifest and annotation data from the R-packages. “IlluminaHumanMethylation450kmanifest” [27] and “IlluminaHumanMethylation450kanno.ilmn12.hg19” [28] (Appendix A).

### 2.5. Validation of Global Methylation Patterns in an Independent Series

With the aim of finding out whether global methylation in regions referred to CpG island is consistent with that from a different cohort, we analyzed data from a previously described CRC series [22], which used the same DNA methylation microarray platform as in the present study. As histological information was not available from this cohort, and given the fact that most SAC are MSS [10], the assumption was made that group CIMP-H1/MSS+ was enriched in SAC; CIMP-H1-H2/MSI+ was enriched in hmMSI-H and the rest of the groups (CIMP-L1-L2-Neg/CIMP-H2/MSS+) in CC.

### 2.6. Oncogene Mutation Status and Microsatellite Instability

We evaluated the mutation status of the *BRAF* and *KRAS* oncogenes using previously described methods [10]. TaqMan probes were used for the detection of *BRAF* V600E mutation. Positive cases were sequenced for exon 15 of *BRAF* as described elsewhere [29]. Mutations in codons 12 and 13 of *KRAS* were assessed by denaturing high-performance liquid chromatography (dHPLC) and confirmed by Sanger sequencing [10].

Microsatellite instability (MSI) was evaluated in the tumor cases using the MSI Analysis System, v.1.2 (Ref: MD1641) (Promega, Madison, WI, USA) as per the manufacturer’s instructions. The cases were classified as MSI-H or microsatellite stable (MSS)/low-level MSI (MSI-L) in agreement with the NCI criteria [30].

### 2.7. Assessment of CpG Island Methylation Phenotype (CIMP)

CIMP was assessed using Methylation-specific multiplex ligation-dependent probe amplification (MS-MLPA) using the CIMP specific SALSA MLPA probemix (Ref: ME042-C1) (MRC-Holland, Amsterdam, The Netherlands). A hundred nanograms of DNA from all 81 cases, were denatured in Tris-EDTA buffer (5 µL total volume) and fragment analysis of fragments were carried out on a capillary sequencer (ABI 3130, Applied Biosystems, Foster City, CA, USA). As normal reference DNA from normal colon mucosa was used. The methylation status of each position was determined using Coffalyser.netTM software, with default settings, and all quality control parameters being within satisfactory range. Inter sample normalization was performed against multiple runs of the reference sample, and additionally, intra-sample normalization was carried out by adjusting all probes to reference probes within each sample. The ratios of peak height of digested vs. undigested sample were compared individually to score the methylation of all probes using Coffalyser. Incompletely methylated genes were considered as methylated. The Ogino et al. [31] 6/8 (CIMP(O)) and Weisenberger et al. [32] (CIMP(W)) criteria for CIMP status assessment were used for further analysis. Briefly, ref. [31]. consider samples to be CIMP-negative when none or less than 6 genes were methylated and CIMP-High when >6 genes were methylated. In contrast, ref. [32] considered 5 genes (*CACNA1G*, *IGF2*, *NEUROG1*, *RUNX3*, *SOCS1*) and the difference between CIMP-Negative vs. CIMP-High was ≤3 vs. >3 methylated genes, respectively.

### 2.8. Statistical Analysis

Statistical analyses were performed using the SPSS package (Version 15.0, Chicago, IL, USA). After the preliminary statistical studies, we conclude that our samples did not hold the assumptions for parametric tests (e.g., non-normality/skewness and small group sizes), as such we leveraged non-parametric tests: Mann-Whitney’s U and Kruskal–Wallis’H to detect significant differences in medians between groups. Associations between categorical variables were tested through the Fisher–Freeman–Halton perfect test for independence. Associations between continuous variables were tested through Spearman’s correlation coefficient. Array data were analyzed with the minfi (1) R-package (2), applying quantile normalization to M-values and *t*-tests to evaluate methylation differences between normal and tumor samples for each genome region. The beta-value has more intuitive biological interpretation, but the M-value is more statistically valid for the differential analysis of methylation levels [33]. The M value is the log_2_ ratio of the intensities of methylated versus unmethylated probes and is calculated using the formula:$$M_{i}=\log_{2}\left(\frac{\max\left(\mathrm{yi},\mathrm{methy},0\right)+\alpha }{\max\left(\mathrm{yi},\mathrm{unmethy},0\right)+\alpha }\right)$$
where yi,methy and yi,unmethy are intensities measured by the ith methylated and unmethylated probes. Alpha is a constant recommended by Illumina, and its default value is 100.

## 3. Results

### 3.1. Global Methylation Levels in Normal Tissue and Their Comparison with Tumoral Tissue

We found no associations between adjacent normal mucosa methylation scores and demographic features (Table 2). When examining the correlations between methylation at differing distances from TSS in normal tissue, associations were found only between adjacent sequence regions, i.e., 250 bp and 1 Kb methylation scores correlated (Spearman’s r = 0.542, *p* = 0.001), as did 1 and 2 Kb methylation (Spearman’s r = 0.883, *p* < 0.001). The methylome scores at 250 bp, 1 kb, and 2 kb did not differ in normal mucosa by adjacent CRC subtype (Table 2).

Comparison between tumor vs. normal tissue revealed that methylation at 1–250 bp was higher in tumor compared to normal mucosa (median difference = 0.0038, *p* = 0.0100) and the opposite was true at 250 bp–1 Kb (median difference = −0.0118, *p* < 0.001) and remarkably at 1–2 Kb, where tumor was strongly more hypomethylated compared to normal tissue (median difference = −0.0402, *p* < 0.001) (Figure 1 and Appendix A). When restricting the analysis to specific histological CRC subtypes, significant differences were found between normal and tumoral tissue. In CC, differences were found at longer distances from the TSS (i.e., 250 bp–1 Kb; *p* < 0.001 and 1 Kb–2 Kb; *p* < 0.001), whereas in hmMSI-H CRC, differences in DNA methylation were predominantly confined to regions in close proximity to the TSS (1–250 bp; *p* = 0.005). SAC showing an intermediate methylation pattern with normal vs. tumoral differences in the methylation scores for all distance ranges from the TSS (Figure 1, Appendix A).

Next, we examined methylation in relation to proximity to a CpG island. Hypermethylation of tumor tissue was strongest at CpG islands (*p* = 2.85 × 10^−12^) and decreased as one moves away from them, becoming hypomethylated in (N/S) shelf (*p* = 2.3 × 10^−6^ y 6.13 × 10^−6^) and Open Sea (*p* = 1.15 × 10^−7^) (Figure 2). Appendix A shows these differences are dependent on the histological CRC subtype. Differences amongst CRC subtypes at CpG islands did not reach statistical significance, although the methylation level at shores, shelves and open sea were significantly different between CC and hmMSI-H. Appendix A shows that the shift from hypermethylation to hypomethylation in our cancer series occurs in a short region of 4 Kb from the CpG island. When comparing with adjacent normal tissue, tumor tissues were significantly less methylated at either shelf, but no statistical differences were observed when analyzing shore regions (N: *p* = 0.335; S: *p* = 0.481) (Appendix A). Moreover, hypermethylation of tumor samples was observed in the upstream regions such as 1st exon and 5′UTR but not at the gene body (Appendix A).

Interestingly, N_shores were significantly more methylated than S_shores (*p* = 0.003), but this difference was not observed between N_ and S_shelf regions (*p* = 0.404) (Appendix A). Validation in an independent CRC cohort showed that cases assigned to the SAC group displayed an intermediate global methylation. However, the difference between the SAC- and hmMSI-H-enriched groups did not reach statistical difference, probably due to the limited number of cases (12 each). Interestingly, the main differences observed amongst groups were found at the island and N_and S_shore locations but not in shelves or open sea sequences (Appendix A).

### 3.2. Relationship between Methylation Scores and Tumoral Tissue Features

In general terms, tumor tissues from hmMSI-H showed higher methylation scores whereas CC showed the lowest scores at all sequence location ranges (Figure 3). When comparing the methylation scores of these tumor histological subtypes amongst each other, significant differences were found between CC and hmMSI-H at all ranges (1–250 bp, 250 bp–1 Kb, 1–2 Kb) and between SAC and hmMSI-H at 250 bp–1 Kb and 1 Kb–2 Kb. Table 3 shows that no associations were found between methylation scores and tumor features except for tumor histology.

### 3.3. Relationship between Methylation Scores and Molecular Markers (MSI, KRAS, BRAF, CIMP)

Those cases displaying MSI-H (*n* = 14) showed more overall methylation at 250 bp than MSS tumors (*n* = 68) (median difference = 0.0053 *p* = 0.025), whereas no significant differences were found at 1–2 Kb or when comparing score values of KRAS mutated versus KRAS wild-type cases. BRAF mutated cases showed a slightly higher score at 250 b; however, they did not reach statistical significance (Table 4). As expected, only methylation at 250 bp, and not at 1 kB or 2 Kb, was significantly associated with CIMP status of tumoral cases. This relationship was observed regardless of the Weisenberger et al. or Ogino et al. criteria used (Table 4). Interestingly, only the methylation status of some of the genes included in the CIMP panel were significantly associated with the global methylation score at 250 bp (CDKN2A (median difference (methylated–unmethylated) = 0.0056; *p* = 0.011), and MLH1 (median difference= 0.006; *p* = 0.026) (Appendix A) whereas no such association was observed for the rest (*CACNA1G*, *CRABP1*, *IGF2*, *NEUROG1*, *RUNX3* and *SOCS1*) (Figure 4).

## 4. Discussion

Altered CpG methylation patterns play a causative role in colorectal carcinogenesis. However, a representation of the complex methylation pattern of tumors using manageable parameters is still missing. Focal DNA hypermethylation, which overlaps with promotors and CpG islands [34,35], is generally assessed by analyzing the DNA methylation status of 5–8 genes (CIMP). Global hypomethylation appears to play a role in carcinogenesis but has no clear marker for measurement. The analysis of hypomethylated regions in cancer has been concentrated in repetitive DNA elements such as Alu and LINE-1, which are implicated in chromosome breakage and chromosome instability [36]. There has been limited analysis of hypomethylated regulatory elements [37], though there are unique sequences also involved in cancer-associated hypomethylation [38]. Epigenetic changes are too global to be reduced to the analysis of few DNA sequences. In this study, three scores were calculated representing the methylation status of three different regions defined by their distance from the transcription start site (250 bp, 1 Kb, 2 Kb) of the associated gene, thus including areas of promoter hypermethylation and global hypomethylation.

Analysis of the normal mucosa revealed a continuum in the methylation patterns as methylation at 1 Kb correlated with that at 250 bp and 2 Kb whilst methylation at 250 bp was not associated with that at 2 Kb.

Our study showed that whilst normal mucosa was more methylated than tumoral mucosa at the 2 and 1 Kb, the opposite was observed for 250 bp. This indicates that the shift between promoter hypermethylation and global hypomethylation is generally found between 250 bp and 1 Kb from TSS and suggests that DNA methylation dynamics affect adjacent and close regions, possibly regulating different targets. Similar results were found when the regions of reference were established considering the CpG island with the shift from hyper to hypomethylation occurring in the transition from island to shelf regions. This finding is in agreement with the work by Liu et al., who showed that, when comparing sessile serrated adenomas with vs. without dysplasia, hypomethylation occurred in concert with hypermethylation, increasing progressively away from the island regions [38]. Here we observed that the methylation scores of adjacent regions (i.e., 250 bp with 1 Kb and 1 Kb with 2 Kb) correlate with each other but do not follow a symmetrical pattern at both sides of the CpG island as N_shores seem to be more methylated than S_shores in CRC tissue compared to adjacent normal mucosa. This may be due to differing distributions of regulatory elements on the north and south sides of CpG islands. Intriguingly, the differences in methylation scores between tumor and normal mucosa are higher in the hypomethylated regions than in promoter-associated hypermethylated regions.

When considering the different histological subtypes, our results suggest that promoter hypermethylation and global genome hypomethylation can be seen as independent molecular pathological mechanisms with a more global hypomethylated pattern in CC, a more focal hypermethylated pattern in hmMSI-H and a mixed phenotype in SAC. These different methylation profiles may contribute to the distinct histological features observed in these CRC subtypes. Other clinical and pathological features were not found associated with methylation scores except for a slight increase in methylation at the 250 bp-1 Kb range in mucin-producing tumors, which have been related with CIMP in previous studies (reviewed in [39]). Our findings are in line those reported by Visone et al., who, using eight CRC stem cell lines from primary CRC, noticed a preponderance of the differentially methylated positions associated with MSI tumors, which do not reside at CpG islands but spread to shelf and open sea regions [40].

The relationship between overall methylation at 1–250 bp and the global methylation status of CpG islands is evidenced by the finding that these genomic regions are the only regions significantly associated with CIMP and MSI status. This reinforces the difference in methylation patterns between 250 bp and 1 Kb. Another implication of these results is that despite the Ogino et al. criteria considering more loci than the Weisenberger et al. panel (8 vs. 5), they both correlate similarly to methylation at 250 bp. Notably, both Weisenberger et al. and Ogino et al. criteria consider in a similar way the genes included in the CIMP to categorize tumors as CIMP high or CIMP low. However, here we show that the contribution to global methylation at 250 bp differs among these genes, and only two genes (*CDKN2A* and *MLH1*) out of eight from the CIMP panel correlated with methylation at 250 bp.

## 5. Conclusions

This study describes the topography of methylation in CRC, highlighting the contribution of hypomethylation in regions from 250 bp to 2 Kb from TSS. Since the main differences in tumor-associated hypomethylation are observed outside the CpG island, the study of non-coding sequences in these locations may be important as these non-coding RNAs could serve as promising diagnostic and prognostic markers in CRC.

## Figures and Tables

**Figure 1 cancers-13-05165-f001:**
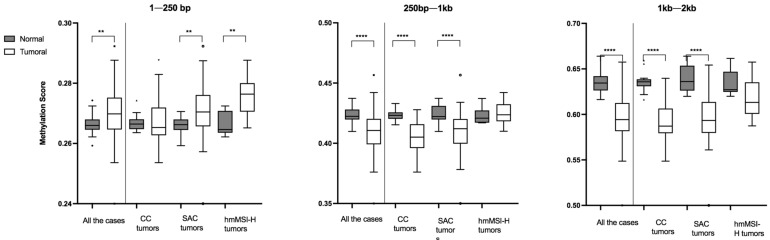
Comparison of global methylation scores between normal and tumor specimens for each tumor histological subtype. CC: conventional carcinoma, SAC: serrated adenocarcinoma, hmMSI-H: colorectal cancer showing histological and molecular features of high level of microsatellite instability, ** *p* < 0.01, **** *p* < 0.0001.

**Figure 2 cancers-13-05165-f002:**
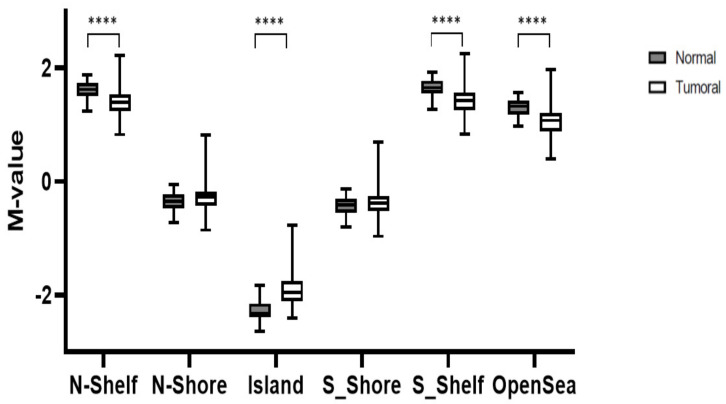
Comparison between normal and tumoral methylation in different locations with respect to CpG island; i.e., (N/S) shores, (N/S) shelves and open sea. **** *p* < 0.0001.

**Figure 3 cancers-13-05165-f003:**
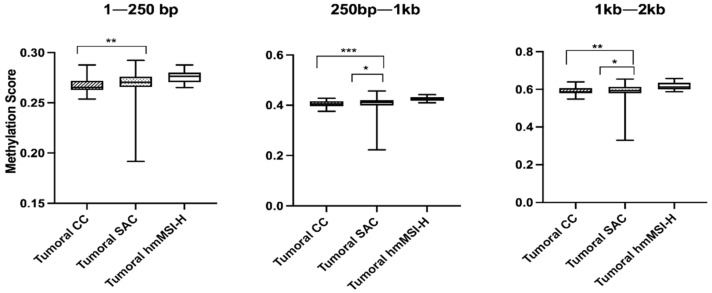
Global methylation scores for each tumor histological subtype. CC: conventional carcinoma, SAC: serrated adenocarcinoma, hmMSI-H: colorectal cancer showing histological and molecular features of high level of microsatellite instability. * *p* < 0.05, ** *p* < 0.01, *** *p* < 0.001.

**Figure 4 cancers-13-05165-f004:**
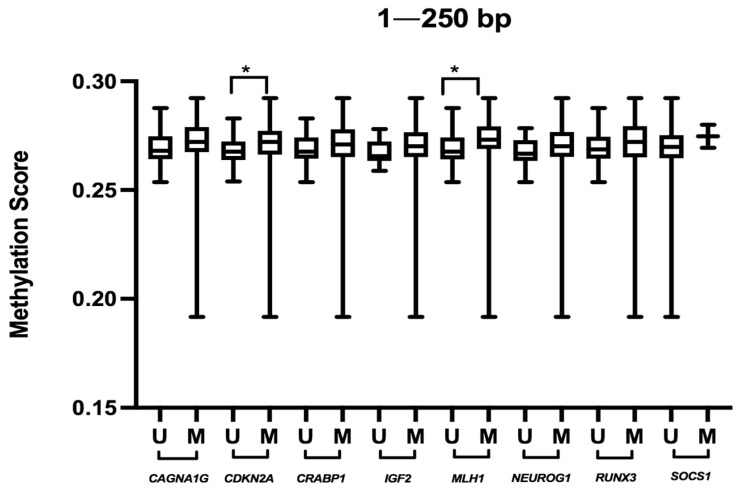
Global methylation score at 1–250 bp according to the methylation status of genes included in the CpG island methylation phenotype (CIMP) panel. U: unmethylated, M: methylated. * *p* < 0.05.

**Table 1 cancers-13-05165-t001:** Demographic, pathological and molecular characteristics of the study population according to tumor histology.

Variable		CC (*n* = 32)	SAC (*n* = 40)	hmMSI-H (*n* = 10)	*p-*Value
Sex	Female	15 (46.9%)	21 (52.5%)	8 (80%)	0.191
	Male	17 (53.1%)	19 (47.5%)	2 (20%)	
Age	≤71	15 (46.9%)	17 (42.5%)	4 (40%)	0.952
	≥72	17 (53.1%)	23 (57.5%)	6 (60%)	
Location	Proximal	19 (59.4%)	28 (70%)	10 (100%)	0.038
	Distal	13 (40.6%)	12 (30%)	0 (0%)	
T	Tis	1 (3.1%)	0 (0%)	0 (0%)	0.453
	T2	3 (9.4%)	8 (20%)	0 (0%)	
	T3	21 (65.6%)	23 (57.5%)	6 (60%)	
	T4	7 (21.9%)	9 (22.5%)	4 (40%)	
N	N0	19 (59.4%)	18 (45%)	4 (40%)	0.720
	N1	5 (15.6%)	9 (22.5%)	2 (20%)	
	N2	8 (25.0%)	8 (25%)	4 (40%)	
M	M0	28 (90.3%)	31 (81.6%)	6 (60%)	0.101
	M1	3 (9.7%)	7 (18.4%)	4 (40%)	
Grade	LG	30 (93.8%)	30 (75%)	7 (70%)	0.050
	HG	2 (6.3%)	10 (25%)	3 (30%)	
Tumor budding	LG	20 (80.3%)	12 (54.5%)	7 (70%)	0.116
	HG	4 (16.7%)	10 (54.5%)	3 (30%)	
Tumor budding No.	Absent	10 (41.7%)	8 (36.4%)	7 (70%)	0.029
	<10	11 (45.8%)	4 (18.2%)	0 (0%)	
	10–19	2 (8.3%)	5 (22.7%)	1 (10%)	
	>20	1 (4.2%)	5 (22.7%)	2 (20%)	
Mucin	Absent	18 (75%)	3 (13.6%)	4 (40%)	<0.001 *
	<50%	5 (20.8%)	7 (31.8%)	3 (3%)	
	50–75%	0 (0%)	2 (9.1%)	2 (20%)	
	>75%	1 (4.2%)	10 (45.5%)	1 (10%)	
*BRAF*	WT	31 (96.9%)	28 (70%)	6 (60%)	0.003 *
	Mutated	1 (3.1%)	12 (30%)	4 (40%)	
*KRAS*	WT	17 (53.1%)	25 (62.5%)	8 (80%)	0.335
	Mutated	15 (46.9%)	15 (37.5%)	2 (20%)	
MSI	MSS/MSI-L	32 (100%)	33 (82.5%)	3 (30%)	<0.001 *
	MSI-H	0 (0%)	7 (17.5%)	7 (70%)	
CIMP Weisemberger et al.	Low CIMP	31 (96.9%)	25 (65.8%)	3 (30%)	<0.001 *
	High CIMP	1 (3.1%)	13 (34.2%)	7 (70%)	
CIMP Ogino et al.	Low CIMP	32 (100%)	29 (76.3%)	3 (30%)	<0.001 *
	High CIMP	0 (0%)	9 (23.7%)	7 (70%)	

Note: Fisher–Freeman–Halton perfect test was applied for testing the ≥2 variate associations of each variable by type of tumor. The Bonferroni correction for multiple comparisons was applied. Associations with type of tumor are considered significant (marked with *) if *p* < 0.05/15 = 0.0033. CC: conventional carcinoma, SAC: serrated adenocarcinoma, hmMSI-H: colorectal cancer showing histological and molecular features of high level of microsatellite instability, MSI: microsatellite instability, MSS: microsatellite stability, MSI-L: low level of MSI, MSI-H: high level of MSI, CIMP: CpG island methylation phenotype.

**Table 2 cancers-13-05165-t002:** Associations between global methylation scores of normal adjacent mucosa and demographic and clinic-pathological features of the study patients.

Variable			250 bp			1 kb			2 kb		
		*n*	Median	IQR	*p*	Median	IQR	*p*	Median	IQR	*p*
All		35	0.266	0.003		0.422	0.008		0.634	0.016	
Age	≤71	14	0.267	0.004	1.000	0.424	0.009	0.678	0.638	0.016	0.263
	>72	21	0.266	0.004		0.422	0.009		0.634	0.022	
Sex	Female	17	0.267	0.005	0.184	0.422	0.010	0.335	0.631	0.017	0.245
	Male	18	0.265	0.004		0.423	0.008		0.637	0.016	
Location	Proximal	22	0.265	0.003	0.335	0.422	0.011	0.335	0.634	0.022	1.000
	Distal	13	0.268	0.005		0.424	0.007		0.636	0.012	
Type	CC	14	0.266	0.003	0.775	0.423	0.005	0.805	0.636	0.008	0.752
	SAC	15	0.266	0.004		0.422	0.011		0.636	0.027	
	hmMSI-H	6	0.265	0.007		0.421	0.010		0.627	0.022	

Note: *p*-values for Mann–Whitney’s *U* test and Kruskal–Wallis’ *H*. The Bonferroni correction for multiple comparisons was applied for each site; *p*-values < 0.05/5 = 0.01 are considered to indicate a significant difference between medians. IQR: interquartile range. CC: conventional carcinoma, SAC: serrated adenocarcinoma, hmMSI-H: colorectal cancer showing histological and molecular features of high level of microsatellite instability.

**Table 3 cancers-13-05165-t003:** Association of global methylation scores in tumor specimens with demographic, clinic-pathological and histologic features of the study patients.

Variable			250 bp			1 kb			2 kb		
		*n*	Median	IQR	*p*	Median	IQR	*p*	Median	IQR	*p*
All		82	0.2698	0.0106		0.4106	0.0211		0.5943	0.0309	
Type	CC	32	0.2653	0.0090	0.0060	0.4051	0.1980	0.0010 *	0.5871	0.0271	0.0120
	SAC	40	0.2704	0.0104		0.4121	0.0205		0.5933	0.0347	
	hmMSI-H	10	0.2763	0.0094		0.4236	0.1410		0.6131	0.0347	
Nationality	Spanish	56	0.2698	0.0106	0.6320	0.4098	0.0207	0.9440	0.5956	0.0308	0.6320
	Finnish	26	0.2698	0.0116		0.4121	0.0239		0.5915	0.0294	
Age	≤71	36	0.2703	0.0137	0.8740	0.4090	0.2530	0.9630	0.5933	0.3140	0.8010
	≥72	46	0.2694	0.0094		0.4106	0.1870		0.5955	0.3140	
Sex	female	44	0.2695	0.0103	0.6220	0.4065	0.2277	0.4080	0.5921	0.0293	0.5330
	Male	38	0.2707	0.0111		0.4136	0.0207		0.5959	0.0302	
Location	Proximal	57	0.2701	0.0107	0.6980	0.4095	0.2200	0.3020	0.5939	0.0333	0.2880
	Distal	25	0.2695	0.0109		0.4163	0.2190		0.5961	0.2090	
T	Tis	1									
	T2	11	0.2701	0.1510	0.3770	0.4040	0.0222	0.1620	0.5969	0.0271	0.0510
	T3	50	0.2700	0.0960		0.4106	0.0222		0.5893	0.0303	
	T4	20	0.2700	0.0137		0.4151	0.2340		0.6048	0.2790	
N	N0	41	0.2695	0.1040	0.9870	0.4040	0.0256	0.8010	0.5939	0.3580	0.9660
	N1	16	0.2708	0.0091		0.4113	0.0124		0.5955	0.0210	
	N2	25	0.2692	0.0140		0.4139	0.0230		0.5927	0.0295	
M	M0	65	0.2694	0.0109	0.4960	0.4078	0.0226	0.1780	0.5939	0.0308	0.2130
	M1	14	0.2704	0.0119		0.4164	0.0177		0.6013	0.0351	
Grade	LG	67	0.2695	0.0102	0.8530	0.4092	0.0205	0.5450	0.5939	0.0304	0.5770
	HG	15	0.2701	0.0143		0.4163	0.0255		0.6023	0.0356	
Tumor budding	LG	39	0.2687	0.0124	0.8530	0.4092	0.0207	0.5450	0.5939	0.0252	0.5770
	HG	17	0.2709	0.0098		0.4166	0.0414		0.6021	0.0435	
Mucin	Absent	26	0.2654	0.0120	0.2820	0.4034	0.0199	0.0320	0.5864	0.0320	0.1070
	Present	15	0.2709	0.0093		0.4159	0.0197		0.6019	0.0299	

Note: The Bonferroni correction for multiple comparisons was applied for each site; *p*-values < 0.05/11 = 0.0045 were considered statistically significant and marked with an asterisk (*).

**Table 4 cancers-13-05165-t004:** Associations of global methylation scores and molecular features of the colorectal carcinomas of the study patients.

Variable			250 bp			1 kb			2 kb		
		*n*	Median	IQR	*p*	Median	IQR	*p*	Median	IQR	*p*
*BRAF*	WT	65	0.2690	0.0104	0.1720	0.4092	0.0203	0.2060	0.0254	0.0270	0.2800
	Mutated	17	0.2731	0.0143		0.4166	0.0254		0.6023	0.0474	
*KRAS*	WT	50	0.2702	0.0097	0.9550	0.4093	0.0226	0.9920	0.5950	0.0376	0.5880
	Mutated	32	0.2691	0.0126		0.4114	0.0184		0.5913	0.0254	
MSI	MSS/MSI-L	68	0.2686	0.0103	0.0250	0.4093	0.0222	0.1150	0.5915	0.0317	0.1430
	MSI-H	14	0.2739	0.0111		0.4155	0.0256		0.6027	0.0313	
CIMP Weisemberger et al.	Low CIMP	59	0.2681	0.0105	0.0380	0.4079	0.0222	0.3170	0.5902	0.0320	0.4410
	High CIMP	21	0.2731	0.0108		0.4152	0.0201		0.5974	0.0282	
CIMP Ogino et al.	Low CIMP	64	0.2681	0.0100	0.0300	0.4093	0.0661	0.4070	0.5915	0.0313	0.8290
	High CIMP	16	0.2739	0.0108		0.4155	0.0203		0.5972	0.0305	

Note: The Bonferroni correction for multiple comparisons was applied for each site; *p*-values < 0.05/5 = 0.01 were considered statistically significant.

## Data Availability

The data set supporting the results of this article are available in the GEO repository, GSE68060 in https://www.ncbi.nlm.nih.gov/geo/query/acc.cgi?acc=GSE68060.

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
