# Peer review of "Global Methylome Scores Correlate with Histological Subtypes of Colorectal Carcinoma and Show Different Associations with Common Clinical and Molecular Features"

_cancers, 2021, doi:10.3390/cancers13205165_

Round 1

Reviewer 1 Report

Page3: There should be an proved IRB protocol number listed here.

Page4: There are a lot of abbreviation used in the paper. Author should consider to use a full Abbr. list, keep them consistent and do not need to repeat under every table. And also, I see mixed use of MSI-H and hmMSI-H, which I suppose they are the same.

        Missing the detail condition and concentration for tissue disruption.

        The only catalog number in the whole paper. Please add Cata# also for the chip and other kits.

        Please indicate the software version of R.

Page6: Add reference of “Ogino et al.” and “Weisemberger et al.”

Figure 6, Please replace with high-resolution picture and adjust the font.

Author Response

-Please see the attached file.

Author response (AR): Thank you for including the attached file. That makes revision easier.

-Page3: There should be an proved IRB protocol number listed here.

AR: The protocol number has now been added in the text.  

-Page4: There are a lot of abbreviation used in the paper. Author should consider to use a full Abbr. list, keep them consistent and do not need to repeat under every table. And also, I see mixed use of MSI-H and hmMSI-H, which I suppose they are the same.

AR: An abbreviation list has now been incorporated in the manuscript. In order to present tables as self-explanatory, we have kept the abbreviations. We leave at the journal policy the decision of using both, one or the other. MSI-H refers to a molecular phenomenon whereas hmMSI-H corresponds to a molecular and histological entity. Therefore, MSI-H and hmMSI-H are not exactly the same as there are some colorectal carcinomas (CRC) with microsatellite instability (MSI-H) which do not show typical histological features associated with microsatellite unstable CRC (i.e.) such as signet-ring cell, mucinous, and medullary carcinoma, intra- and peritumoral infiltrating lymphocytes, “Crohn-like” inflammatory reaction, poor differentiation, tumor heterogeneity and “pushing” tumor border These features are typically found in hmMSI-H.

-Missing the detail condition and concentration for tissue disruption.

AR: Protocols details are now provided

-The only catalog number in the whole paper. Please add Cata# also for the chip and other kits.

AR: References of the kits and reagents used in the study are now provided in material and methods

 -Please indicate the software version of R.

AR: The R software version has now been included.

 -Page6: Add reference of “Ogino et al.” and “Weisemberger et al.”

 AR: All references to Ogino or Weisenberger have now been followed by “et al”

-Figure 6, Please replace with high-resolution picture and adjust the font.

AR: Figure 6 has been deleted in the manuscript following the editor and a reviewer´s suggestion

Reviewer 2 Report

This is a retrospective analysis of 82 cases of colon cancer that correlates the clinical and pathological features to different methylation patterns.

The work is quite interesting and coherent as regards the part relating to the correlation between methylation and pathological characteristics.

Instead, the correlation between methylation patterns and survival is totally inappropriate, for the following reasons:

- The study population is small

- The analysis includes patients in different stages, in particular metastatic and non-metastatic patients, so survival should be stratified at least by stage, but also by the main known prognostic factors (sites of metastases, CEA, RAS, BRAF, MSI …)

- Furthermore, the prognosis could be influenced by the treatments received by the patients, in particular if metastatic, which are not reported.

Survival analysis must be removed from both the results and the conclusions.

Author Response

This is a retrospective analysis of 82 cases of colon cancer that correlates the clinical and pathological features to different methylation patterns.

The work is quite interesting and coherent as regards the part relating to the correlation between methylation and pathological characteristics.

Instead, the correlation between methylation patterns and survival is totally inappropriate, for the following reasons:

- The study population is small

- The analysis includes patients in different stages, in particular metastatic and non-metastatic patients, so survival should be stratified at least by stage, but also by the main known prognostic factors (sites of metastases, CEA, RAS, BRAF, MSI …)

- Furthermore, the prognosis could be influenced by the treatments received by the patients, in particular if metastatic, which are not reported.

Survival analysis must be removed from both the results and the conclusions.

Author response (AR): We appreciate that the reviewer found our study interesting. Following his/her suggestion and understanding his/her concerns about it we have deleted the survival part from the manuscript.

Reviewer 3 Report

The authors performed interesting analyses trying to describe the methylome in CRC and the difference comparing distinct pathological subtypes. I find this part of the paper very interesting. It is well performed and described. However, the associations to survival have several limitations that actually makes it very difficult to interpret these results and I will suggest the authors to skip this part.

I miss data on the clinical handling of the patients. How were they treated? Neoadjuvant chemotherapy? Surgical procedures? Adjuvant chemotherapy? And most importantly differences between the groups?

Why was survival analyses performed on 51 random patients and not the entire cohort? There does not seem to be a reasonable explanation for this selection?

Did the authors test for interaction in the multiple cox regression analyses. Several of the parameters are known to be highly linked and this may affect performance of this analyses.

Why did the authors include patients with all stages (I-IV)? This will likely impact on the results, especially the survival analyses. It also makes the interpretation very difficult as these patients have been treated differently (palliative chemotherapy, versus adjuvant chemotherapy, versus no chemotherapy?). A cohort with all stage II or III would have been a more solid set-up. Numbers are very small in the individual survival analyses. What is measured? Overall survival, disease free survival?

Follow-up is approximately half a year. It is a bit unclear how old the cohorts are (when they were operated) but in the MM section the authors refer to papers published back in 2010. Therefore, I guess the cohorts are rather old then. Why only half a year of follow-up then? The majority of patients should be dead now. In the present form, these survival data are far from mature. Why group in two? Why not used continuous data in a cox analyses and take advantages of all the biological variation in the dataset?

It is a bit unclear to begin with what the results from table 2 represents. Associations. So the median listed in the table is what exactly? This is explained later. Difference between tumor and normal. Then in table 3 (with no normal tissue to compare with) what does the median refer to here? In addition, the single p-value in the situation with three listed medians. Is it the statistical difference between all three at once, or just two of them?

The authors present a very large number of comparisons, and p-values. Did they consider to test for multiple comparisons?

Figure two is very illustrative.

Author Response

The authors performed interesting analyses trying to describe the methylome in CRC and the difference comparing distinct pathological subtypes. I find this part of the paper very interesting. It is well performed and described. However, the associations to survival have several limitations that actually makes it very difficult to interpret these results and I will suggest the authors to skip this part.

-I miss data on the clinical handling of the patients. How were they treated? Neoadjuvant chemotherapy? Surgical procedures? Adjuvant chemotherapy? And most importantly differences between the groups?

Why was survival analyses performed on 51 random patients and not the entire cohort? There does not seem to be a reasonable explanation for this selection?

Did the authors test for interaction in the multiple cox regression analyses. Several of the parameters are known to be highly linked and this may affect performance of this analyses.

Why did the authors include patients with all stages (I-IV)? This will likely impact on the results, especially the survival analyses. It also makes the interpretation very difficult as these patients have been treated differently (palliative chemotherapy, versus adjuvant chemotherapy, versus no chemotherapy?). A cohort with all stage II or III would have been a more solid set-up. Numbers are very small in the individual survival analyses. What is measured? Overall survival, disease free survival?

Follow-up is approximately half a year. It is a bit unclear how old the cohorts are (when they were operated) but in the MM section the authors refer to papers published back in 2010. Therefore, I guess the cohorts are rather old then. Why only half a year of follow-up then? The majority of patients should be dead now. In the present form, these survival data are far from mature. Why group in two? Why not used continuous data in a cox analyses and take advantages of all the biological variation in the dataset?

AR: We understand the reviewer´s concerns about the survival analysis and following his/her suggestion and that of the editor we have removed the survival analysis from the manuscript.

-It is a bit unclear to begin with what the results from table 2 represents. Associations. So the median listed in the table is what exactly? This is explained later. Difference between tumor and normal. Then in table 3 (with no normal tissue to compare with) what does the median refer to here? In addition, the single p-value in the situation with three listed medians. Is it the statistical difference between all three at once, or just two of them?

AR: The median values represent that of the methylation score in normal adjacent samples, as indicated in the figure heading; they are not established from a comparison between tumor vs. normal samples. Table 3 is the equivalent table but with methylation scores obtained in tumor specimens. The p-value associated with three listed medians is the statistical different between all three at once.

-The authors present a very large number of comparisons, and p-values. Did they consider to test for multiple comparisons?

AR: Thanks for this comment. In the revised manuscript, we have now applied the Bonferroni correction for multiple comparison. See the corrected levels in all the tables.

 -Figure two is very illustrative.

AR: Thank you your fruitful and enriching comments. Additionally, English language has been revised by a native English speaker (LJF)

Reviewer 4 Report

In this manuscript, Turpin-Sevilla et al. revealed a difference in the “global methylation pattern” among histological subtypes of colorectal cancers (CRCs). The authors performed a DNA methylation assay to assess the global methylation levels among three different subtypes: serrated adenocarcinoma, conventional carcinoma, and hmMSI-high carcinoma. While this manuscript is interesting, I have concerns about the data presented in it.

My core concerns are as follows:

  1. The classification of CRCs should follow the latest WHO classifications. Detailed information about conventional carcinoma (adenocarcinoma, NOS or including specific subtypes?) should be clearly stated in the manuscript because the authors mentioned the production of mucin, which can indicate differentiation. 
  2. The minimum number for applying the Chi-squared test should be considered. 
  3. In figure 2, what kind of histological subtypes affected the difference in the M-value? Related to this point, in figure S5, the histological differences are not clear. The purely histological features should be considered. 
  4. In figure 3, is this methylation pattern specific to CRCs? Or is it specific to some histological subtype? If this pattern is universal and just confirms the quality of this experiment, this figure would be supplementary. 
  5. How about the mRNA expression of highly methylated candidates, such as CDKN2A and MLH1? This point should be considered.
  6. Regarding survival analysis, what is the conclusion? The data not showing statistically significant differences could be supplementary information. 

Minor points:

  1. The format of the manuscript should be adjusted to that of Cancers.  
  2. Table 1: what is the relation between “Tumor budding” and “Tumor budding No.”?
  3. Figure 1: the labeling of the Y-axis should be “Methylation score.”
  4. In the methods section 2.5, the authors mentioned PI3KCA,but it is not mentioned in the results section. 
  5. Page 4 line 8: three of the “mm3” instances should be superscripts.
  6. Page 4 line 13: “cat no” should be “Cat. No.”.
  7. A half-space is needed before and after the inequality throughout the manuscript. 
  8. Page 5 line 46, “ul” should be “µl”.
  9. In figure 6, the characters of the label are too small.  

Author Response

In this manuscript, Turpin-Sevilla et al. revealed a difference in the “global methylation pattern” among histological subtypes of colorectal cancers (CRCs). The authors performed a DNA methylation assay to assess the global methylation levels among three different subtypes: serrated adenocarcinoma, conventional carcinoma, and hmMSI-high carcinoma. While this manuscript is interesting, I have concerns about the data presented in it.

My core concerns are as follows:

  1. The classification of CRCs should follow the latest WHO classifications. Detailed information about conventional carcinoma (adenocarcinoma, NOS or including specific subtypes?) should be clearly stated in the manuscript because the authors mentioned the production of mucin, which can indicate differentiation. 

Author response (AR): The reviewer´s comment is pertinent and, in order to clarify the correlation between the study groups and the WHO classification, the following two paragraphs have now been added in Material and Methods:

“CC group comprises the WHO entities of adenocarcinoma NOS, mucinous and non-mucinous. No adenoma-like, micropapillary,adenosquamous adenocarcinomas, or signet ring cell, poorly cohesive, undifferentiated sarcomatoid carcinomas were included in our CC series [WHO blue book 5th edition reference] (….)

“In addition, a previous series of 10 CRCs [10,20] showing MSI-H molecular and histological features (mucinous, and medullary carcinoma, intra- and peritumoral infiltrating lymphocytes, “Crohn-like” inflammatory reaction, poor differentiation, tumor heterogeneity and “pushing” tumor border [8]) termed hmMSI-H and corresponding to the WHO medullary adenocarcinoma subtype [WHO blue book 5th edition reference]”

  1. The minimum number for applying the Chi-squared test should be considered. 

AR: The reviewer is correct that, when some cells have small sample sizes, other statistical tests are recommended (such as Fisher’s exact test). However, this is feasible with variables having 2 levels each (i.e., contingency tables of dimension 2x2). In our Table 1, all the associations tested involved more than 2 levels in at least one of the variables. Therefore, Fisher’s test cannot be applied. Furthermore, previous literature has shown that Chi-squared test is quite robust even with very small expected cell frequencies (Camilli & Hopkins, 1978; Delucchi, 1983)

Camilli, G., & Hopkins, K. D. (1978). Applicability of chi-square to 2 × 2 contingency tables with small expected cell frequencies. Psychological Bulletin, 85(1), 163–167. https://doi.org/10.1037/0033-2909.85.1.163

Delucchi, K. L. (1983). The use and misuse of chi-square: Lewis and Burke revisited. Psychological Bulletin, 94, 166–176. https://doi.org/10.1037/0033-2909.94.1.166

  1. In figure 2, what kind of histological subtypes affected the difference in the M-value? Related to this point, in figure S5, the histological differences are not clear. The purely histological features should be considered. 

AR: In figure 2 all CRC cases were considered. Figure S5 aims to validate our results in an independent series which used the same microarray platform for methylome assessment. As histological information was not available from this cohort, and given the fact that most SAC are MSS, it was taken the assumption that group CIMP-H1/MSS+ was enriched in SAC; CIMP-H1-H2/MSI+ was enriched in hmMSI-H and the rest of groups (CIMP-L1-L2-Neg/CIMP-H2/MSS+) in CC. Nevertheless, in order to answer the reviewer´s question on what kind of histological subtypes affected the difference in the M-value we have now included a new figure (Figure S2) showing the differences in M-value according to histological diagnoses from our series.

  1. In figure 3, is this methylation pattern specific to CRCs? Or is it specific to some histological subtype? If this pattern is universal and just confirms the quality of this experiment, this figure would be supplementary. 

AR: The methylation pattern reflected in Figure 3 is that of the CRC from our series. In order to make that clear the figure legend has been rephrased as follows:

“Differential methylation schema (tumoral CRC M-value – normal adjacent colorectal mucosa M-value) showing the tumor-associated hyper- and hypomethylation according to the genome locations with respect to the CpG island”

Following the reviewer´s suggestion this figure has been relocated as supplementary.

  1. How about the mRNA expression of highly methylated candidates, such as CDKN2A and MLH1? This point should be considered.

AR: CDKN2A and MLH1 are amongst the most studied genes for their promoter methylation, especially in CRC. The link between hypermethylation and lower mRNA have been demonstrated in early studies [Trzeciak L, et al. Cancer Lett. 2001;163:17-23; Arnold CN, et al. Int J Cancer. 2003;106:66-73] and we did not consider as a novel finding to demonstrate this association. 

  1. Regarding survival analysis, what is the conclusion? The data not showing statistically significant differences could be supplementary information. 

AR: Following the editor and reviewers´ comments the survival analysis has been removed from the manuscript

Minor points:

  1. The format of the manuscript should be adjusted to that of Cancers.  

AR: The manuscript has been written following the Cancers manuscript template.

  1. Table 1: what is the relation between “Tumor budding” and “Tumor budding No.”?

AR: The following text in Material and Methods has been added to explain the difference:

“As explained elsewhere, tumor budding was considered as low when presented less than 10 tumor buds and high grade tumor budding when showing 10 or more tumor buds”

  1. Figure 1: the labeling of the Y-axis should be “Methylation score.”

AR: The reviewer´s observation is correct and the Y-axes of Figure 1 have been renamed accordingly

  1. In the methods section 2.5, the authors mentioned PI3KCA,but it is not mentioned in the results section. 

AR: PIK3CA information was incomplete. Therefore, the mention to PIK3CA has been removed from the manuscript.

  1. Page 4 line 8: three of the “mm3” instances should be superscripts.

AR: This correction has been made

  1. Page 4 line 13: “cat no” should be “Cat. No.”

AR: The reference to kits used in the study have been uniformly presented as “Ref:” 

  1. A half-space is needed before and after the inequality throughout the manuscript. 

AR: A minimum of half-space has been granted before and after each paragraph.

  1. Page 5 line 46, “ul” should be “µl”.

AR: This correction has been made

  1. In figure 6, the characters of the label are too small.  

AR: Figure 6 has been deleted in the manuscript following the editor and a reviewer´s suggestion

Round 2

Reviewer 2 Report

The paper is suitable for publication after the revision

Author Response

Reviewer 2 has said that "The paper is suitable for publication after the revision" and no other comments have been found in his/her revision. Therefore, no further actions are required at this point by the authors. 

Reviewer 3 Report

With survival analyses removed I find the paper much more trustworthy.

Author Response

Authors´ response (AR): The survival analysis has been removed from the manuscript and the English language has been revised by our colleague Lochlan Fennell who, apart from being a native English speaker, has a deep knowledge on the topics covered in the mansucript.

Reviewer 4 Report

Almost all concerns were well addressed. However, regarding "#2. the Chi-squared test", Fisher's exact test can apply for more than 2x2 tables or  I think authors can divide the samples into two groups with the optimal threshold. Otherwise, the authors should discuss the limitations of the conclusion with the Chi-squared test in the present study. 

Author Response

Reviewer´s comment: Almost all concerns were well addressed. However, regarding "#2. the Chi-squared test", Fisher's exact test can apply for more than 2x2 tables or  I think authors can divide the samples into two groups with the optimal threshold. Otherwise, the authors should discuss the limitations of the conclusion with the Chi-squared test in the present study.

Authors´ response: Thanks for this comment. The reviewer is right, Fisher's exact test was generalized to tables of any dimension by Freeman & Hamilton (1951). This exact test is available in SPSS.

In the revised manuscript, we replaced the Chi-squared results reported in Table 1 with the corresponding p-values derived from the Fisher-Freeman-Hamilton exact test. Therefore, the limited sample size in some cells of Table 1 is not a problem in the revised version. Note that this potential problem did not affect any other part of the manuscript.  Freeman, G. H., & Halton, J. H. (1951). Note on an Exact Treatment of Contingency, Goodness of Fit and Other Problems of Significance. Biometrika38(1/2), 141–149. https://doi.org/10.2307/2332323

Therefore, p-values in Table 1, where Chi-squared was originally applied, have been changed, accordingly and the table note has been modified as follows:

"Note: Fisher-Freeman-Halton perfect test was applied for testing the ≤2 variate associations of each variable by type of tumor"

Likewise, the Material and Method section dedicated to statistical analysis has been chaged as follows:

"Associations between categorical variables were tested through the Fisher-Freeman-Halton perfect test for independence"